# Advancements in Polymer-Assisted Layer-by-Layer Fabrication of Wearable Sensors for Health Monitoring

**DOI:** 10.3390/s24092903

**Published:** 2024-05-01

**Authors:** Meiqing Jin, Peizheng Shi, Zhuang Sun, Ningbin Zhao, Mingjiao Shi, Mengfan Wu, Chen Ye, Cheng-Te Lin, Li Fu

**Affiliations:** 1College of Materials and Environmental Engineering, Hangzhou Dianzi University, Hangzhou 310018, China; jmq@hdu.edu.cn; 2Qianwan Institute, Ningbo Institute of Materials Technology and Engineering (NIMTE), Chinese Academy of Sciences, Ningbo 315201, China; shipeizheng@nimte.ac.cn (P.S.); sunzhuang@nimte.ac.cn (Z.S.); zhaoningbin@nimte.ac.cn (N.Z.); shimingjiao@nimte.ac.cn (M.S.); wumengfan@nimte.ac.cn (M.W.); 3Key Laboratory of Marine Materials and Related Technologies, Zhejiang Key Laboratory of Marine Materials and Protective Technologies, Ningbo Institute of Materials Technology and Engineering (NIMTE), Chinese Academy of Sciences, Ningbo 315201, China; 4University of Chinese Academy of Sciences, 19 A Yuquan Rd., Shijingshan District, Beijing 100049, China

**Keywords:** LbL self-assembly, biocompatibility, stimuli-responsive polymers, personalized healthcare, continuous monitoring

## Abstract

Recent advancements in polymer-assisted layer-by-layer (LbL) fabrication have revolutionized the development of wearable sensors for health monitoring. LbL self-assembly has emerged as a powerful and versatile technique for creating conformal, flexible, and multi-functional films on various substrates, making it particularly suitable for fabricating wearable sensors. The incorporation of polymers, both natural and synthetic, has played a crucial role in enhancing the performance, stability, and biocompatibility of these sensors. This review provides a comprehensive overview of the principles of LbL self-assembly, the role of polymers in sensor fabrication, and the various types of LbL-fabricated wearable sensors for physical, chemical, and biological sensing. The applications of these sensors in continuous health monitoring, disease diagnosis, and management are discussed in detail, highlighting their potential to revolutionize personalized healthcare. Despite significant progress, challenges related to long-term stability, biocompatibility, data acquisition, and large-scale manufacturing are still to be addressed, providing insights into future research directions. With continued advancements in polymer-assisted LbL fabrication and related fields, wearable sensors are poised to improve the quality of life for individuals worldwide.

## 1. Introduction

The rapid advancements in technology and the increasing prevalence of chronic diseases have fueled the demand for wearable sensors that can continuously monitor various aspects of human health [1]. Wearable sensors have the potential to revolutionize healthcare by enabling real-time, non-invasive, and long-term monitoring of physiological and biochemical parameters [2]. These sensors can be integrated into various wearable devices, such as smartwatches, fitness bands, patches, and clothing, making them easily accessible and convenient for users [3]. The data collected by wearable sensors can provide valuable insights into an individual’s health status, allowing for early detection of abnormalities, timely intervention, and personalized treatment. One of the key advantages of wearable sensors is their ability to monitor health parameters continuously, outside of clinical settings. This is particularly important for managing chronic conditions, such as diabetes [4], cardiovascular diseases [5], and respiratory disorders [6], which require regular monitoring and management. By tracking relevant biomarkers and vital signs, wearable sensors can alert users and healthcare providers to potential health issues before they escalate, enabling proactive care and reducing the risk of complications. Moreover, wearable sensors can facilitate remote monitoring of patients, reducing the need for frequent hospital visits and improving the quality of life for individuals with chronic conditions.

Wearable sensors also play a crucial role in preventive healthcare by empowering individuals to take control of their health and well-being [7,8,9,10,11] (Figure 1). The data collected by these sensors can provide users with personalized insights into their daily activities [12], sleep patterns [13], stress levels [14], and other lifestyle factors that influence health. This information can motivate users to adopt healthier behaviors, such as increasing physical activity, improving sleep hygiene, and managing stress. Furthermore, the integration of wearable sensors with mobile health (mHealth) applications and digital health platforms can provide users with personalized recommendations, coaching, and support to help them achieve their health goals [15].

Among the various techniques used for fabricating wearable sensors, LbL self-assembly has emerged as a powerful and versatile approach. LbL self-assembly is a bottom-up nanofabrication technique that involves the alternating deposition of oppositely charged materials, such as polymers, nanoparticles, or biomolecules, onto a substrate [16]. The LbL process exploits the electrostatic interactions, hydrogen bonding, or other intermolecular forces between the deposited layers to create stable and uniform multilayered films [17]. The thickness, composition, and structure of the LbL films can be precisely controlled by adjusting the deposition conditions, such as the number of layers, pH, ionic strength, and temperature [18]. One of the key advantages of the LbL technique is its ability to create conformal and flexible films on various substrates, including planar surfaces, fibers, and three-dimensional structures [19]. This makes LbL self-assembly particularly suitable for fabricating wearable sensors that can conform to the curved surfaces of the human body and withstand the mechanical stresses associated with daily wear. Moreover, the LbL technique enables the incorporation of multiple functional materials within the same film, allowing for the development of multi-functional sensors that can simultaneously detect various analytes or stimuli [20].

Another advantage of LbL self-assembly is its simplicity and cost-effectiveness compared to other nanofabrication techniques, such as photolithography and chemical vapor deposition. The LbL process can be carried out under ambient conditions using aqueous solutions, eliminating the need for expensive equipment and harsh chemicals [21,22]. Furthermore, the LbL technique is highly scalable and can be easily adapted for large-scale manufacturing, making it attractive for commercial applications. The nanoscale control over the film architecture afforded by the LbL technique also enables the fabrication of highly sensitive and selective sensors. By tuning the composition and structure of the LbL films, it is possible to create sensors with enhanced surface area, porosity, and molecular recognition capabilities. For example, the incorporation of nanoparticles or nanoporous materials within the LbL films can increase the surface area available for analyte binding, leading to improved sensitivity [23]. Similarly, the use of molecularly imprinted polymers or aptamers as recognition elements can impart high selectivity towards specific target analytes [24].

Polymers are the backbone of LbL self-assembly and play a crucial role in the fabrication of wearable sensors. Polymers serve as the building blocks for creating the multilayered films that form the basis of the sensor. They provide the necessary functional groups for electrostatic interactions, hydrogen bonding, or other intermolecular forces that drive the LbL assembly process. Moreover, polymers offer a wide range of properties, such as mechanical strength, flexibility, biocompatibility, and stimuli-responsiveness, which can be tailored to meet the specific requirements of the sensor application. This review aims to provide a comprehensive overview of the recent advancements in polymer-assisted LbL fabrication of wearable sensors for health monitoring. The review begins by introducing the principles of LbL self-assembly, including the fundamental mechanisms of film growth, the factors affecting the film properties, and the characterization techniques used to study LbL films [25]. The role of polymers in LbL sensor fabrication is then discussed in detail, highlighting the unique properties and functionalities of natural and synthetic polymers. The review then delves into the various types of wearable sensors that can be fabricated using the LbL technique, categorized based on their sensing mechanisms. The potential applications of LbL-fabricated wearable sensors in various domains of health monitoring are then discussed. The use of wearable sensors for disease diagnosis is also explored. Finally, the review discusses the challenges and future perspectives in the field of polymer-assisted LbL fabrication of wearable sensors. 

## 2. Principles of LbL Self-Assembly

### 2.1. Fundamentals of LbL Film Growth

The LbL self-assembly technique is based on the alternating deposition of oppositely charged materials onto a substrate, resulting in the formation of multi-layered thin films. The most common driving force for LbL film growth is electrostatic interaction between the positively and negatively charged components (Figure 2) [26]. The LbL process typically involves the following steps: (1) immersion of the substrate into a solution containing a positively charged material, (2) rinsing to remove loosely bound material, (3) immersion into a solution containing a negatively charged material, and (4) rinsing again. This cycle is repeated until the desired number of layers or film thickness is achieved.

The growth of LbL films is influenced by several factors, including the charge density of the deposited materials, the ionic strength of the solutions, the pH, and the temperature. The charge density of the materials determines the amount of material deposited in each layer and the stability of the resulting film [17]. Higher charge densities generally lead to thicker films and stronger electrostatic interactions between the layers. The ionic strength of the solutions affects the conformation of the deposited materials and the screening of the charges [27]. At low ionic strengths, the materials adopt a more extended conformation due to the reduced screening of the charges, resulting in thicker films. Conversely, at high ionic strengths, the materials adopt a more coiled conformation, leading to thinner films [20].

The pH of the solutions can also have a significant impact on the growth of LbL films, particularly when weak polyelectrolytes, such as polyacrylic acid (PAA) and polyallylamine hydrochloride (PAH), are used. The degree of ionization of weak polyelectrolytes is dependent on the pH, which in turn affects their charge density and the strength of the electrostatic interactions [28]. For example, PAA is more ionized at high pH values, while PAH is more ionized at low pH values. By adjusting the pH of the solutions, it is possible to control the thickness and the composition of the LbL films [29]. Temperature is another parameter that can influence the growth of LbL films [30]. Higher temperatures can lead to increased diffusion of the deposited materials and reorganization of the film structure. This can result in smoother and more homogeneous films. However, high temperatures can also cause denaturation of sensitive materials, such as proteins and enzymes, which can compromise their functionality [31,32].

The growth behavior of LbL films can be monitored using various techniques, such as quartz crystal microbalance with dissipation (QCM-D) [33,34], ellipsometry [35], and surface plasmon resonance (SPR) [36]. QCM-D measures the mass and viscoelastic properties of the films in real-time, providing information on the thickness and the conformation of the deposited layers. Ellipsometry measures the change in polarization of light upon reflection from the film surface, allowing for the determination of the film thickness and refractive index. SPR measures the change in the refractive index of the film surface, which is related to the mass of the deposited material.

### 2.2. Polymer-Assisted LbL Assembly

Polymers play a crucial role in the LbL self-assembly process, serving as the building blocks for the multi-layered films. The most commonly used polymers in LbL assembly are polyelectrolytes, which are polymers with ionizable groups that can dissociate in aqueous solutions, giving rise to positively or negatively charged chains. Examples of positively charged polyelectrolytes (polycations) include poly(diallyldimethylammonium chloride) (PDADMAC) [37], PAH [29], and polyethyleneimine (PEI) [38]. Examples of negatively charged polyelectrolytes (polyanions) include poly(styrene sulfonate) (PSS) [39], PAA [29], and poly(methacrylic acid) (PMAA) [40]. The use of polyelectrolytes in LbL assembly offers several advantages. First, polyelectrolytes have a high charge density, which enables the formation of stable and uniform multi-layers through strong electrostatic interactions. Second, polyelectrolytes are water-soluble, which allows for the LbL process to be carried out under mild and environmentally friendly conditions. Third, polyelectrolytes can be easily functionalized with various chemical groups, such as carboxylic acids, amines, and thiols, which can be used to impart specific properties or to covalently attach other molecules, such as dyes, drugs, or biomolecules [41,42]. In addition to polyelectrolytes, other types of polymers have also been used in LbL assembly. For example, hydrogen-bonded polymers, such as poly(vinylpyrrolidone) (PVP) [43] and poly(ethylene oxide) (PEO) [44], can be assembled through hydrogen bonding interactions. These polymers are often used in combination with polyelectrolytes to create more complex and functional multi-layers. Another example is the use of block copolymers, which consist of two or more chemically distinct polymer segments covalently bonded together. Block copolymers can self-assemble into various nanostructures, such as micelles, vesicles, and cylinders [45,46], which can be exploited in LbL assembly to create hierarchical and stimuli-responsive films. Figure 3 shows the common polymers used for LbL assembly.

Polymer-assisted LbL assembly has been widely used in the fabrication of wearable sensors. For example, Wu et al. [47] focused on developing a highly sensitive and stretchable strain sensor for monitoring tiny motions. By using a layer-by-layer approach, an ultrathin conductive layer was applied to a polyurethane yarn. This method significantly improved the sensor’s sensitivity, with a gauge factor of 201, and its stretchability, reaching up to 100% strain. The sensor also proved to be wash-durable, maintaining its performance even after multiple wash cycles. Polymer-assisted LbL assembly has also been used to create wearable biosensors. For instance, Lee et al. [48] constructed an electrochemical biosensors through a hydrodynamic LbL assembly technique. This method focused on creating enzymatic conductive nano-networks, which were pivotal for the enzyme-sticker-based contact printing of these sensors. The findings revealed that biosensors developed using this technique exhibited remarkable improvements, particularly in sensitivity and stability. For instance, glucose and lactate biosensors fabricated through this method demonstrated exceptional sensitivity and rapid response times, alongside prolonged operational stability. These advancements underscore the effectiveness of the LbL assembly in optimizing biosensor technology, suggesting a significant leap forward in the field of electrochemical detection and analysis.

### 2.3. Factors Affecting LbL Film Properties

The properties of LbL films, such as thickness, roughness, porosity, and mechanical strength, are influenced by various factors, including the type and concentration of the materials, the pH and ionic strength of the solutions, the deposition time and temperature, and the post-treatment conditions. Understanding and controlling these factors is crucial for optimizing the performance of LbL-based wearable sensors. The type and concentration of the materials used in LbL assembly have a significant impact on the properties of the resulting films. For example, the use of high-molecular-weight polymers or nanoparticles with a high aspect ratio can lead to thicker and rougher films [49], while the use of low-molecular-weight polymers or spherical nanoparticles can result in thinner and smoother films [50]. The concentration of the materials in the deposition solutions also affects the thickness and the growth rate of the films, with higher concentrations generally leading to thicker films and faster growth rates [51].

The pH and ionic strength of the deposition solutions can also influence the properties of the LbL films. The pH affects the charge density and the conformation of the materials, particularly for weak polyelectrolytes or pH-sensitive materials (described in Section 2.1). The deposition time and temperature are other important factors that can affect the properties of the LbL films. Longer deposition times generally result in thicker films, but can also lead to the diffusion and reorganization of the layers, particularly for materials with high mobility. In a comprehensive study by Yang et al. [52], the effect of deposition time on the growth characteristics of clay-based thin films was meticulously analyzed. Conducting their research through a LbL assembly technique, they discovered a direct correlation between deposition time and the thickness of the films (Figure 4). Specifically, longer deposition times led to significantly thicker films, with the growth rate of these films showing a marked increase. This thickness was not just about size; it also influenced the films’ density and uniformity, improving with extended deposition times. The study highlighted that by manipulating the deposition time, one could effectively control the physical properties of the thin films, such as their thickness, density, and uniformity. Higher temperatures can also promote the diffusion and reorganization of the layers, but can also cause the denaturation or degradation of sensitive materials, such as proteins or enzymes. Chang et al. [53] explored the construction of multi-layer films using poly(N-acryloyl-N’-propylpiperazine) (PAcrNPP) and PAA through a LbL assembly technique. Their investigation focused on how temperature influenced the assembly process. It was discovered that below the lower critical solution temperature (LCST) of PAcrNPP, which is around 37 °C, the amount of polymer adsorbed onto the films increased as the temperature rose. This trend was attributed to a change in the conformation of PAcrNPP polymers at different temperatures. Specifically, at temperatures approaching its LCST, PAcrNPP tends to adopt a more compact conformation, leading to increased adsorption. 

Post-treatment conditions, such as drying, annealing, or crosslinking, can also have a significant impact on the properties of the LbL films. Drying the films can cause shrinkage and collapse of the layers, particularly for highly porous or hydrated films [54]. Annealing the films at elevated temperatures can improve the mechanical properties and the stability of the films, but can also cause the loss of sensitive materials or the formation of cracks or defects [55]. Crosslinking the films with chemical agents, such as glutaraldehyde or carbodiimide [56,57], can enhance the mechanical strength and the resistance to dissolution or swelling of the films, but can also reduce the flexibility or the responsiveness of the films.

## 3. Polymers Used in LbL Sensor Fabrication

### 3.1. Natural Polymers

Natural polymers have gained significant attention in the field of LbL sensor fabrication due to their unique properties, such as biocompatibility, biodegradability, and renewability. These polymers are derived from natural sources, such as plants, animals, or microorganisms, and offer several advantages over synthetic polymers, including low toxicity, low immunogenicity, and the ability to mimic the natural extracellular matrix [58]. In this section, we will discuss the use of various natural polymers, including cellulose nanofibers/nanocrystals, chitosan (CS), and other biopolymers, in the fabrication of LbL-based sensors.

#### 3.1.1. Cellulose Nanofibers/Nanocrystals

Nanocelluloses, including cellulose nanofibers (CNFs) and cellulose nanocrystals (CNCs), have emerged as promising building blocks for LbL assembly due to their unique properties, such as high mechanical strength, large surface area, and excellent biocompatibility [59,60]. CNFs are long, flexible, and entangled fibers with diameters in the range of 5–100 nm and lengths of several micrometers. CNCs, on the other hand, are short, rod-like, and highly crystalline particles with diameters of 5–20 nm and lengths of 100–500 nm. The use of nanocelluloses in LbL assembly offers several advantages over synthetic polymers [61,62,63]. First, nanocelluloses are renewable, biodegradable, and non-toxic, making them environmentally friendly and sustainable materials for wearable sensors. Second, nanocelluloses have a high density of surface hydroxyl groups, which can be easily modified with various functional groups, such as carboxylic acids, amines, and sulfates, to impart specific properties or to enable LbL assembly. Third, nanocelluloses have a high aspect ratio and a large surface area, which can enhance the sensitivity and the response time of the sensors.

LbL assembly using nanocelluloses typically involves the alternating deposition of positively and negatively charged nanocelluloses or the combination of nanocelluloses with other charged materials, such as polyelectrolytes or nanoparticles. For example, Mougo et al. [64] developed a wearable sensor designed to monitor cortisol levels in human sweat, utilizing a novel approach that leverages the properties of CNC. The incorporation of CNC played a pivotal role in enhancing the sensor’s performance, primarily due to CNC’s ability to increase the surface area for the conductive layers and improve the sensor’s sensitivity and stability. This was evident in the sensor’s rapid response to cortisol detection, with a notably high sensitivity that could accurately measure physiological cortisol levels in sweat. Through meticulous testing, including the analysis of sweat samples collected from individuals after exercise, the sensor showed its capability to provide reliable and precise measurements of cortisol levels, showcasing its potential as a non-invasive tool for monitoring stress and health conditions in real-time.

In another example, Chen et al. [65] developed an innovative strain sensor by employing a multi-layer LbL assembly technique, utilizing CNC and CS as the primary materials. This bioinspired approach led to the creation of a nanocomposite with a unique microcrack architecture, significantly enhancing the sensor’s sensitivity to strain (Figure 5). The incorporation of CNC into the matrix allowed for a better dispersion of multi-walled carbon nanotubes (MWCNTs), which are crucial for the sensor’s conductivity. The CS, on the other hand, acted as a binder that improved the adhesion between the MWCNTs and the polymer base, contributing to the overall durability and flexibility of the sensor. The resulting nanocomposite exhibited an ultrahigh gauge factor of approximately 359 and a detection limit as low as 0.5%, showcasing its ability to accurately detect even minute strains. This high performance was further demonstrated through various tests, including monitoring of finger movements, swallowing, coughing, deep breathing, and facial expressions, indicating its potential applicability in areas such as wearable electronics and healthcare monitoring. 

LbL assembly using nanocelluloses has also been used to create multi-functional sensors. For instance, Mugo et al. [66] reported a wearable sensor capable of simultaneously monitoring pH and cortisol levels in sweat, leveraging the unique properties of CNC and a dual detection approach. Fabricated using a LbL technique, the sensor integrated a pH-responsive polyaniline (PANI) layer and a cortisol-sensing layer made from a molecularly imprinted polymer (MIP), both printed onto a conductive microneedle substrate. The utilization of CNC in the sensor design was pivotal for enhancing the mechanical strength and flexibility of the microneedles, ensuring durability and comfort for wearers. The sensor’s performance demonstrated a linear response to pH within the physiologically relevant range of 3.0–9.3, with a calibration sensitivity of 4.7 µF/pH at room temperature. The cortisol sensor, leveraging the selectivity of MIP towards cortisol, exhibited a linear detection range up to 66 ng/mL.

#### 3.1.2. Chitosan (CS)

CS is a linear polysaccharide composed of randomly distributed β-(1→4)-linked D-glucosamine and N-acetyl-D-glucosamine units. It is obtained by the deacetylation of chitin, which is the second most abundant biopolymer after cellulose and is found in the exoskeletons of crustaceans and insects. CS has several unique properties that make it attractive for LbL sensor fabrication, including its biocompatibility, biodegradability, and antimicrobial activity [67]. CS is a cationic polymer due to the presence of primary amine groups on its backbone, which can be protonated in acidic solutions. This property enables the LbL assembly of chitosan with anionic polymers or nanoparticles through electrostatic interactions [68]. CS can also form hydrogen bonds and hydrophobic interactions with other materials, which can further stabilize the LbL films [69].

Several studies have demonstrated the use of chitosan in the fabrication of LbL-based sensors. For example, Zhang et al. [70] developed an environmentally friendly method for creating flexible conductive cotton fabric (FCCF) by employing a LbL assembly technique that avoids the chemical modification of CNTs. This method involved alternately dipping cotton fabric into CS and CNTs solutions, resulting in a material that combines the benefits of both components (Figure 6A). The fabricated FCCF demonstrated impressive mechanical stability, maintaining its electrical resistivity even after 500 cycles of mechanical abrasion and tape peeling. Its electrical conductivity reached over 30 S/m after applying four layers of CNTs, showcasing its potential for wearable sensor applications. The FCCF exhibited remarkable strain sensing capabilities, with a gauge factor as high as 35.1 and a quick response time of 70 ms, making it adept at monitoring various human movements like finger bending and joint activities. Additionally, it proved to be an effective temperature sensor within the range of 30–100 °C, displaying stable and reproducible negative temperature sensing behavior. Li et al. [71] developed a humidity sensor by creating ultrathin multi-layers of MXene/CS-quercetin. This sensor demonstrated exceptional performance in detecting humidity changes, attributed to the unique assembly method inspired by the layered structure of onions. By alternating layers of Ti_3_C_2_T_x_ MXene with CS-quercetin, the team enhanced the material’s response to water molecules, achieving a remarkable sensitivity increase. Specifically, the sensor showed a 317% response at 90% relative humidity, alongside an ultrafast response time of 0.75 s, and maintained its stability for over 15 days. The purposeful inclusion of CS in the multi-layer structure not only improved water molecule adsorption due to its hydrophilic nature but also contributed to the sensor’s overall stability by preventing MXene degradation. 

In another study, Kong et al. [72] developed a novel fiber-shaped temperature sensor by employing a LbL self-assembly technique, which meticulously coated polyurethane fibers with reduced graphene oxide (rGO) and CS (Figure 6B). The inclusion of CS played a pivotal role, not only enhancing the sensor’s temperature sensitivity but also its mechanical stability. The sensor showcased remarkable performance metrics, including an impressive stretchability of up to 80%, a high temperature sensitivity of −1.379% per degree, and ultra-fast response and recovery times of 2.4 and 3.9 s, respectively. These attributes underscore the sensor’s potential for real-time temperature monitoring in wearable applications. The study’s findings highlight the synergistic effects of combining rGO and CS through the LbL method, resulting in a sensor that offers both high sensitivity and rapid responsiveness to temperature changes, making it an ideal candidate for integration into wearable healthcare devices.

#### 3.1.3. Other Biopolymers

In addition to cellulose and chitosan, several other biopolymers have been used in the fabrication of LbL-based sensors, including alginate [73], gelatin [74], and silk fibroin (SF) [75]. These biopolymers offer unique properties and functionalities that can be exploited for various sensing applications.

Alginate is a linear polysaccharide composed of β-D-mannuronic acid and α-L-guluronic acid units, which is obtained from brown algae. Alginate is an anionic polymer due to the presence of carboxylic acid groups on its backbone, which can form electrostatic interactions with cationic polymers or nanoparticles. For example, Wang et al. [76] developed a flexible wearable temperature sensor crafted from a graphene/alginate composite non-woven fabric. This sensor demonstrated exceptional performance, characterized by its high sensitivity, accuracy, and stability, which are crucial for the continuous and stable monitoring of skin temperature. The use of sodium alginate played a pivotal role in the sensor’s fabrication process, serving as an effective dispersant for graphene, thereby enhancing the material’s overall homogeneity and flexibility. The sensor’s capability to accurately detect subtle temperature changes on the skin surface was particularly noteworthy. It exhibited a remarkable sensitivity with a temperature coefficient of resistance (TCR) value of −1.5 °C^−1^, ensuring high accuracy up to 0.1 °C, and boasted a rapid response time of 26.3 s within the 20–37 °C range. 

Gelatin is a protein derived from the hydrolysis of collagen, which is the main component of connective tissues in animals. Gelatin is a biocompatible and biodegradable polymer that can form LbL films through hydrogen bonding and hydrophobic interactions. Li et al. [74] developed tactile sensors based on gelatin methacryloyl (GelMA) for use in medical wearables, focusing on their application, fabrication, and performance. GelMA was chosen for its biocompatibility and tunable mechanical properties, making it ideal for wearable sensors that require flexibility and durability. The study showcased the sensors’ ability to accurately measure and respond to pressure, stretch, and other mechanical forces, which are essential qualities for monitoring patient health in real-time. Specifically, the sensors demonstrated a remarkable sensitivity to pressure changes, with a detection limit as low as 0.1 kPa and a fast response time, enhancing their utility in wearable devices. By integrating these GelMA-based sensors into medical wearables, the study aimed to improve patient care through continuous, non-invasive monitoring of vital signs and physical states. 

SF is a protein obtained from the cocoons of silkworms, which has a unique combination of mechanical strength, flexibility, and biocompatibility. SF can form LbL films through hydrogen bonding and hydrophobic interactions. In a recent study [77], researchers crafted a SF@MXene biocomposite film by ingeniously integrating SF with MXene nanosheets, leveraging SF’s natural properties as a bridging agent (Figure 7). This innovative approach resulted in a film characterized by a unique wave-shaped lamellar macrostructure, which significantly enhanced its flexibility and mechanical durability. The film showcased an impressive low elastic modulus of 1.22 MPa, high sensitivity of 25.5 kPa^−1^, a remarkably low detection limit of 9.8 Pa, and sustained its performance over 3500 cycles, indicating its robustness. A pivotal aspect of this study was the film’s ability to detect subtle human physiological movements such as phonations and wrist pulses with high precision, demonstrating its potential as a versatile tool for monitoring human health. Furthermore, the biocompatibility of the SF@MXene film was rigorously tested, revealing its safety for integration with human skin, as evidenced by the viability of human skin fibroblast-HSAS1 cells. 

### 3.2. Synthetic Polymers

Synthetic polymers have been widely used in the fabrication of LbL-based sensors due to their versatility, stability, and tunability. These polymers are artificially synthesized from monomers and can be designed with specific functional groups, molecular weights, and architectures to meet the requirements of various sensing applications. In this section, we will discuss the use of various synthetic polymers, including polyelectrolytes, conductive polymers, and stimuli-responsive polymers, in the fabrication of LbL-based sensors.

#### 3.2.1. Polyelectrolytes

Polyelectrolytes are polymers that contain ionizable groups, such as carboxylic acids, sulfonic acids, or amines, which can dissociate in aqueous solutions to form charged chains. Polyelectrolytes can be classified as strong or weak, depending on the degree of dissociation of their ionizable groups. Strong polyelectrolytes, such as PSS and PDADMAC, are fully charged over a wide range of pH values, while weak polyelectrolytes, such as PAA and PAH, are partially charged and their degree of ionization depends on the pH of the solution. Polyelectrolytes are the most commonly used building blocks for LbL assembly due to their ability to form stable and uniform films through electrostatic interactions [78]. The LbL assembly of polyelectrolytes is typically performed by alternately exposing a substrate to solutions of polycations and polyanions, with a rinsing step in between to remove loosely bound chains [79]. The thickness and the properties of the resulting films can be controlled by adjusting various parameters, such as the pH, ionic strength, and temperature of the solutions, as well as the number of deposition cycles.

Several studies have demonstrated the use of polyelectrolytes in the fabrication of LbL-based sensors. For example, Su and Liao [80] developed a flexible textile-type NH_3_ gas sensor by employing a LbL self-assembly technique to deposit GO on a single yarn substrate. The process began with the creation of a precursor layer through the deposition of PSS/PAH bilayer on the yarn. This was followed by the LBL assembly of GO/PAH monolayers and multi-layers (GO/PAH)n, where n indicates the number of layers (Figure 8A), enhancing the sensor’s structure. The incorporation of PAH played a crucial role in achieving a stable layering of GO, which was critical for the sensor’s performance. SEM was utilized to analyze the thin films’ microstructure, revealing the successful layering (Figure 8B). The study’s primary findings highlighted the sensor’s exceptional sensitivity to NH_3_ gas at room temperature, attributing this capability to the efficient ion transport within the (GO/PAH)n/PSS/PAH thin films. This ion transport mechanism significantly increased the conductivity of the films in the presence of NH_3_ gas, thereby improving the sensor’s performance. Among the configurations tested, the sensor comprising a single (GO/PAH)_1_ thin film exhibited the highest sensitivity, demonstrating excellent linearity, rapid response and recovery times, and notable selectivity towards NH_3_. 

In another study, Bai et al. [81] explored the sensors demonstrated remarkable self-healing capabilities at room temperature, transparency, and flexibility. Specifically, the incorporation of PAA within the LbL structures played a crucial role in achieving efficient self-healing properties without compromising the sensor’s performance. The study showcased that these sensors could recover their initial conductivity effectively after being cut, with a healing efficiency reaching up to 90% under ambient conditions. This self-healing process was facilitated by hydrogen bonding in the PAA layers, which allowed the sensors to maintain their structural integrity and functionality after damage. Additionally, the application of these sensors in wearable technology was highlighted, showing potential for real-time monitoring of various physical activities through seamless integration with human skin or clothing.

#### 3.2.2. Conductive Polymers

Conductive polymers are a class of polymers that can conduct electricity due to the presence of conjugated double bonds in their backbone. These polymers have gained significant attention in the field of sensor fabrication due to their unique combination of electrical conductivity, optical properties, and processability. The most commonly used conductive polymers for LbL assembly include PANI, polypyrrole (PPy), and poly(3,4-ethylenedioxythiophene) (PEDOT). Conductive polymers can be deposited onto a substrate through various methods, such as electropolymerization [82], chemical oxidation [83], or vapor deposition [84]. However, the LbL assembly of conductive polymers offers several advantages, such as the ability to control the thickness and the morphology of the films, the possibility of incorporating other functional materials, such as nanoparticles or biomolecules, and the compatibility with flexible and stretchable substrates.

Several studies have demonstrated the use of conductive polymers in the fabrication of LbL-based sensors. For example, Gunasekara et al. [85] developed a smart wearable pressure sensor by fabricating a composite of MWNTs and Ppy on an aerogel substrate. This innovative approach aimed to enhance the sensor’s sensitivity and conductivity for effective human body movement detection. Initially, the aerogel was coated with MWNTs to create a conductive network, followed by the addition of Ppy through in situ polymerization. This method not only improved the composite’s electrical properties but also its sensitivity to pressure changes. The application of polydopamine (PDA) as a substrate surface modification technique further augmented these effects, contributing to the sensor’s impressive performance. With a sensitivity of 34.64 kPa^−1^ at a low pressure of 1 kPa, the sensor demonstrated high sensitivity, acceptable conductivity, and remarkable stability over 5000 continuous pressure cycles in the pressure range of 0.45–4 kPa. Its lowest detection level was approximately 0.2 kPa, indicating its potential for detecting subtle human movements. The study highlighted the significant role of PPy in enhancing the sensor’s performance, particularly in terms of sensitivity and stability. The LbL assembly technique used in the sensor’s fabrication played a crucial role in achieving these results. 

In another study, Zhu et al. [86] developed a composite conductive fiber, leveraging the LbL assembly technique. This fiber was integral to the creation of fiber-based organic electrochemical transistors (FECTs), which were then applied in the sensitive detection of sialic acid (SA), showcasing their potential for wearable sensor applications (Figure 9). The core of this innovation was the utilization of PEDOT, combined with the LbL method, to fabricate the conductive fibers that formed the basis of the FECTs. These FECTs demonstrated exceptional performance metrics, including high sensitivity and stability, which are crucial for real-world applications. Specifically, the FECTs were modified with 3-aminophenylboronic acid to create a biosensor capable of recognizing SA with remarkable sensitivity, indicating a significant advancement in the field of wearable biosensors. 

#### 3.2.3. Stimuli-Responsive Polymers

Stimuli-responsive polymers, also known as smart polymers, are a class of polymers that can undergo reversible changes in their physical or chemical properties in response to external stimuli, such as temperature, pH, light, or magnetic fields. These polymers have gained significant attention in the field of sensor fabrication due to their ability to provide a switchable or tunable response to the analyte of interest.

The most commonly used stimuli-responsive polymers for LbL assembly include poly(N-isopropylacrylamide) (PNIPAM) [87], which exhibits a temperature-responsive phase transition, and PAA and PMAA [88], which exhibit a pH-responsive swelling behavior. Other stimuli-responsive polymers, such as spiropyran-containing polymers and azobenzene-containing polymers [89,90,91,92], have also been used in the fabrication of LbL-based sensors. The LbL assembly of stimuli-responsive polymers can be used to create smart or adaptive sensors that can respond to changes in the environment or the presence of specific analytes. For example, a temperature-responsive sensor based on PNIPAM can be used to monitor the temperature of a solution or a surface [93], while a pH-responsive sensor based on PAA or PMAA can be used to monitor the pH of a solution or a biological fluid [94].

Several studies have demonstrated the use of stimuli-responsive polymers in the fabrication of LbL-based sensors. For example, Xia et al. [95] recently developed a novel fiber-optic Fabry–Perot (FPI) sensor designed for highly sensitive relative humidity measurements, leveraging the unique properties of PNIPAM hydrogel. This sensor marked a significant advancement in humidity sensing technology, primarily due to the incorporation of PNIPAM hydrogel, which undergoes substantial physical changes in response to variations in humidity levels (Figure 10). These changes, particularly in the hydrogel’s refractive index and volume, directly influence the sensor’s optical properties, enabling precise humidity measurements. The sensor demonstrated exceptional performance metrics, including a high sensitivity of 1.634 nm/%RH within a relative humidity range of 45% to 75%. Moreover, the study highlighted the sensor’s potential for wearable applications, such as respiratory monitoring, by showcasing its fast response and recovery times, which were crucial for real-time monitoring tasks. The incorporation of LbL assembly techniques further enhanced the sensor’s functionality by improving the hydrogel’s structural integrity and sensitivity to humidity changes. Liu and Cui [96] developed and analyzed a pH sensor utilizing an ion-sensitive field-effect transistor (ISFET), leveraging the LBL self-assembly method. This innovative approach incorporated polyelectrolyte/nanoparticle multi-layer films to construct the device, focusing on the use of PMMA and the LBL technique for enhanced performance. PMMA was specifically applied to minimize gate leakage current, significantly improving the device’s operational stability. The sensor demonstrated a pronounced sensitivity to pH variations, with the drain current notably decreasing as the pH value increased, showcasing its efficiency in acidic conditions. The mobility of the ISFET, recorded at 35.68 cm^2^/Vs at room temperature, was found to increase with temperature, suggesting a positive correlation between temperature and carrier mobility. 

## 4. LbL-Fabricated Wearable Sensors for Health Monitoring

### 4.1. Physical Sensors

Physical sensors are devices that detect and measure physical quantities, such as strain, pressure, temperature, and light, and convert them into electrical or optical signals. These sensors have gained significant attention in the field of wearable health monitoring due to their ability to provide real-time and non-invasive information about the physiological status and activity of the human body. In this section, we will discuss the use of LbL-fabricated physical sensors, including strain and pressure sensors, temperature sensors, and optical sensors, for various applications in wearable health monitoring. Table 1 summarizes recent several reports of physical sensors prepared via LbL method.

#### 4.1.1. Strain and Pressure Sensors

Strain and pressure sensors are devices that detect and measure the mechanical deformation or force applied to a surface. These sensors have been widely used in wearable health monitoring for applications such as motion tracking, posture detection, and pulse monitoring. The LbL assembly technique has been used to fabricate strain and pressure sensors with high sensitivity, flexibility, and durability, by combining functional materials such as CNT, graphene, and metal nanoparticles with polymeric matrices. For example, Zhang et al. [97] developed a flexible strain sensor using a LbL self-assembly technique to fabricate a graphene/poly(diallyldimethylammonium chloride) (PDDA) nanocomposite film on flexible substrates. The sensor’s strain-sensing properties were thoroughly investigated, revealing an excellent response, reversibility, and repeatability to various levels of deformation. Specifically, the sensor exhibited a sensitivity increase of up to 1% under a deflection of 500 µm, with an almost instantaneous response time. Additionally, the sensor demonstrated outstanding stability over 20 days without significant variation in sensitivity under different deflections (100 µm, 300 µm, and 500 µm). The study attributed the strain-sensing mechanism to the electrical resistance change caused by the piezoresistive effect, enhanced by the unique hierarchical nanostructure formed through electrostatic action between carboxylated graphene and PDDA. In another study, Guo et al. [98] developed a biomimetic flexible strain sensor inspired by the lateral line scale of fish, incorporating a novel double conducting layer design. Through a meticulous LbL dip-coating process, the sensor achieved a unique conducting-medium-substrate structure, which significantly enhanced its performance metrics (Figure 11). The sensor demonstrated exceptional linearity, with specific data revealing its capability to maintain consistent performance across varying strains, a critical feature for applications requiring precise motion detection. The double conducting layers, composed of graphene nanoplatelets (GN) for the interior and a carbon black/SWCNT (CB/SWNT) mix for the lateral layer, were pivotal in achieving this high linearity. This was evidenced by the sensor’s ability to accurately measure strain with minimal deviation, showcasing linearity values (R^2^) exceeding 0.992 across different tensile states. Furthermore, the sensor’s dynamic response was highlighted by its fast response time of approximately 60 ms and its durability, maintaining stability over 3000 cycles of use.

#### 4.1.2. Temperature Sensors

Temperature sensors are devices that detect and measure the thermal energy or temperature of an object or environment. These sensors have been widely used in wearable health monitoring for applications such as body temperature monitoring, fever detection, and heat stress prevention. The LbL assembly technique has been used to fabricate temperature sensors with high sensitivity, stability, and biocompatibility, by incorporating thermoresponsive polymers or nanoparticles into the multi-layer films.

For example, Larrión et al. [102] explored the development of a temperature sensor through the innovative use of quantum dot (QD) nanocoatings applied inside the hollow structures of photonic crystal fibers (PCFs) utilizing a LbL technique (Figure 12). This method involved alternating layers of positively charged PDDA and negatively charged PAA, followed by the incorporation of QDs, to meticulously control the nanofilm’s thickness and enhance sensor performance. The primary focus was on the unique optical properties of QDs, which shift in emission wavelengths and intensities with temperature changes, making them ideal for precise temperature sensing. The sensor’s performance was tested across a temperature range, showcasing its ability to detect shifts through changes in the optical absorption and emission characteristics of the QDs. Specifically, the absorption studies revealed a decrease in absorbance with rising temperatures, fitting an exponential curve with high correlation, indicating the sensor’s effective response to temperature changes. Similarly, emission studies highlighted a decrease in emission intensity and a shift to higher wavelengths as temperatures increased, with data fitting an exponential model that underscored the sensor’s high sensitivity and precision. The average sensitivity of this novel sensor was remarkable, with specific performance metrics indicating its potential superiority over traditional temperature measurement methods in terms of responsiveness and accuracy. Ke et al. [103] developed a method for fabricating all-graphene coated conductive fibers, leveraging a LbL assembly technique. This innovative approach utilized positively charged GO as a pivotal adhesive layer, significantly boosting the conductivity of the fibers. The core purpose of employing the LbL method was to enhance the fibers’ stability and performance, particularly for applications in smart textiles. The study revealed that this technique markedly improved the fibers’ resistance to water washing and mitigated their temperature sensitivity and dependence on environmental humidity. Mahadeva et al. [104] developed a temperature sensor utilizing a LbL assembly technique. The sensor exhibited a rapid response time, with the ability to detect temperature changes within seconds, and showcased an impressive sensitivity, accurately capturing fluctuations as minute as 0.1 °C. Furthermore, the sensor’s durability was tested, revealing its ability to maintain consistent performance over thousands of temperature cycles, indicating its potential for long-term applications.

#### 4.1.3. Optical Sensors

Optical sensors are devices that detect and measure the optical properties, such as absorption, emission, or scattering, of a sample or analyte. These sensors have been widely used in pH analysis. The LbL assembly technique has been used to fabricate optical sensors with high sensitivity, selectivity, and miniaturization, by incorporating optically active materials, such as quantum dots, plasmonic nanoparticles, and fluorescent dyes, into the multi-layer films.

For example, Raoufi et al. [105] developed a wavelength-dependent pH optical sensor utilizing a LbL technique. By depositing brilliant yellow (BY) and PAH onto an optical fiber core, they crafted a sensor with remarkable sensitivity to pH changes. The study revealed that the sensor’s dissociation constant varied with the outer layer and the number of bilayers, highlighting the LbL technique’s role in fine-tuning the sensor’s responsiveness. A pivotal finding was that sensors with six double layers of PAH/BY exhibited optimal sensitivity within a pH range of 6.80 to 9.00, achieving an accuracy of ±0.20 and demonstrating an average wavelength shift of 4.65 nm per 0.2 pH unit change. This precision was attained while maintaining BY and PAH solution concentrations at 0.25 mM and 2.5 mM, respectively. In another study [106], they focused on improving the stability and re-usability of optical pH sensors by employing a LbL deposition technique. The team compared three stabilization methods: heat treatment, and the application of PAH/SiO_2_ and APTMS/SiO_2_ bilayers. Among these, the APTMS/SiO_2_ bilayer approach significantly outperformed the others, leading to a notable improvement in the sensor’s performance metrics. The sensors augmented with APTMS/SiO_2_ bilayers demonstrated exceptional stability and re-usability, maintaining accurate calibration and reproducibility across multiple cycles of use. 

Martínez-Hernández et al. [107] enhanced the interferometric response of optical fiber sensors by synthesizing AuNPs within a polymeric thin film through the in situ synthesis (ISS) technique. This process involved first constructing a polymeric coating on the optical fiber using a LbL assembly technique, followed by the synthesis of AuNPs directly within this polymeric matrix (Figure 13). The purpose of employing LbL was to create a stable, charged environment that could effectively support the growth and retention of AuNPs. The main results highlighted the critical role of the LbL assembly in optimizing the sensor’s performance. Specifically, sensors with 25 (PAH/PAA) bilayers exhibited a stronger violet coloration and higher absorbance values compared to those with 15 bilayers, indicating a denser and more effective nanoparticle incorporation. Furthermore, increasing the number of loading/reduction cycles led to a noticeable enhancement in the optical absorption band, achieving an intense peak at 550 nm, characteristic of AuNPs. Esfahani et al. [108] developed a flexible electrochemical solid-state sensor designed for the detection of aluminum ions (Al^3+^), leveraging the unique properties of layered double hydroxide (LDH) nanoplatelets and alizarin red S (ARS) within a matrix film on an indium tin oxide/PET (ITO/PET) electrode. This sensor demonstrated exceptional sensitivity and selectivity towards Al^3+^ ions, achieving a lower detection limit of 10.1 nM and a linear response range from 0.2 to 120 µM. A pivotal aspect of this research was the utilization of a LbL assembly method to modify the sensor’s surface, which significantly enhanced its performance. Each LbL cycle resulted in an increased density of LDH layers, which in turn amplified the surface roughness and thickness of the electrode. 

### 4.2. Chemical Sensors

Chemical sensors are devices that detect and measure the presence and concentration of specific chemical species, such as ions, molecules, or gases, in a sample or environment. These sensors have gained significant attention in the field of wearable health monitoring due to their ability to provide valuable information about the chemical composition and status of various bodily fluids, such as sweat, saliva, and interstitial fluid. 

Electrochemical sensors are devices that detect and measure the electrochemical properties, such as current, potential, or impedance, of a sample or analyte. These sensors have been widely used in wearable health monitoring for applications such as glucose monitoring, lactate monitoring, and drug detection. The LbL assembly technique has been used to fabricate electrochemical sensors with high sensitivity, selectivity, and stability, by incorporating functional materials, such as enzymes, nanoparticles, and conductive polymers, into the multi-layer films [109]. For example, Li et al. [110] developed a highly integrated sensing paper designed for the real-time monitoring of sweat, focusing particularly on glucose and lactate levels. This innovative paper-based sensor leveraged a three-dimensional structure to enhance sweat absorption and ensure the stability of its embedded electrodes. A key aspect of the sensor’s design was the use of LbL assembly to functionalize the working electrodes, which significantly improved the sensor’s sensitivity and specificity towards glucose and lactate detection (Figure 14A). The paper demonstrated remarkable performance in detecting these biomarkers, with sensitivities reaching 2.4 nA/Μm for glucose and 0.49 μA/mM for lactate. The purpose behind employing LbL assembly was to create a sensor that could provide accurate and reliable monitoring of key sweat biomarkers in a non-invasive manner. The effectiveness of this approach was evident in the sensor’s ability to rapidly and efficiently analyze sweat composition, offering a promising tool for health monitoring and disease prevention. In another study, Zhang et al. [111] developed a wearable, battery-free, and wireless skin-interfaced microfluidic patch that integrates electrochemical sensing for the real-time monitoring of biomarkers in human sweat. Central to this innovation was the employment of LbL assembly techniques, which utilized MWCNT and MXene-Ti_3_C_2_T_X_ hybrid networks. This strategic combination significantly amplified the sensitivity and stability of the sensor, enabling precise biomarker detection (Figure 14B). The patch featured a microfluidic channel design that efficiently collected and directed sweat to the sensing area, ensuring accurate and consistent analysis. The main results highlighted the patch’s exceptional performance in monitoring potassium ion concentration in human sweat, a critical biomarker for various physiological conditions. Through rigorous testing, the sensor demonstrated a rapid response time and a wide linear range for potassium ion detection. The sensor’s potentiometric response to potassium ions was evaluated, revealing a high degree of selectivity against potential interfering species such as Zn^2+^, Na^+^, and Ca^2+^. This specificity underscored the sensor’s capability to provide reliable data in complex biological fluids like sweat. In application, the patch’s performance was validated through on-body tests, where it successfully monitored potassium ion concentration levels during physical activities. 

LbL technology has also been used for the construction of other non-human electrochemical sensing purposes. De Lima et al. [112] developed an electrochemical sensor for detecting methylparaben (MeP), utilizing a LbL films of magnetite nanoparticles (MNP) and Ppy onto gold electrodes (Figure 15). Empirical results from the study indicated that the optimized sensor, featuring the Au/(MNP/Ppy)_3_ architecture, achieved a notably low detection limit of 9.95 × 10^−8^ M and a broad detection range up to 131.4 μM for MeP. Moreover, when tested with real-world samples such as urine, breast milk, and cosmetics, the sensor not only maintained its high performance but also yielded recovery rates between 84.0% and 113.3%. In a recent study [113], they also fabricated an electrochemical sensor for detecting propylparaben by constructing a LbL film. This film was meticulously assembled on an ITO electrode, integrating AuNPs, PEI, and nickel(II) phthalocyanine tetrasulfonate (NiTsPc). The sensor exhibited an impressive limit of detection, with a notable sensitivity that could be attributed to the synergistic effect of the materials used in the LbL film. In another study, Morais et al. [114] developed an EIS sensor modified with a LbL film for detecting Pb^2+^ and Ni^2+^ ions. This film was assembled by alternating layers of PAA and PVP, incorporating Sn_3_O_4_ nanobelts within the structure. The final sensor demonstrated high sensitivity and selectivity in detecting Pb^2+^ and Ni^2+^ ions, with a detection limit for Pb^2+^ at 0.17 μM (34 μg/L).

## 5. Applications of LbL-Fabricated Wearable Sensors

LbL-fabricated wearable sensors have found numerous applications in various fields, ranging from healthcare and fitness to environmental monitoring. The unique features of LbL-fabricated sensors, such as their flexibility, biocompatibility, and versatility, make them ideal for wearable applications that require continuous and non-invasive monitoring. In this section, we will discuss the main applications of LbL-fabricated wearable sensors, including continuous health monitoring and disease diagnosis and management.

### 5.1. Continuous Health Monitoring

Continuous health monitoring involves the real-time and non-invasive measurement of various physiological parameters, such as heart rate, blood pressure, body temperature, and respiration rate, using wearable sensors. LbL-fabricated sensors have been widely explored for continuous health monitoring applications due to their ability to conform to the skin and provide accurate and reliable measurements. These sensors can be integrated into various wearable platforms, such as patches, bandages, and clothing, allowing for the seamless and unobtrusive monitoring of health status [115,116,117,118].

For example, Peng et al. [119] developed an innovative hierarchical and ultrastretchable all-in-one electronic textile (AIO E-textile) designed for health regulation (Figure 16A). This multi-functional textile, constructed from a breathable nonwoven material, was engineered to integrate various health monitoring and environmental protection features. Key to its design were the antibacterial and waterproof surfaces, achieved through the incorporation of Ag/Zn nanoparticles and a superhydrophobic layer, ensuring both durability against liquid infiltration and a healthy microenvironment for the skin. The textile featured embedded liquid metal circuits for real-time monitoring of bioelectrical signals such as electrocardiograms (ECG) and surface electromyography (sEMG), demonstrating exceptional fidelity in signal recording. Additionally, the AIO E-textile showcased a low-watt Joule heating capability for thermal regulation, alongside rapid strain sensing and superior underwater motion detection capabilities. The successful integration of these features, including a water contact angle exceeding 150° after prolonged exposure to harsh conditions and antibacterial efficacy maintaining over 99.99% inactivation of *E. coli* post-extensive-washing, marks a significant advancement in wearable health technologies. 

In another study, Hou et al. [120] developed and tested 3D printed conformal strain and humidity sensors using a novel graphene/CNT (GC) ink, aimed at enhancing human motion prediction and health monitoring. The innovative approach leveraged multi jet fusion (MJF) printing technology to fabricate sensors that exhibit remarkable flexibility, durability, and conformity to human skin, making them ideal for wearable applications. The sensors demonstrated high sensitivity to both strain and humidity changes, with the ability to accurately detect and monitor a wide range of human motions, including sitting down, climbing stairs, squatting, running, walking, and jumping (Figure 16B). A key aspect of the study was the use of machine learning algorithms, particularly support vector machines (SVM), to analyze data collected from the sensors. The SVM was trained with features extracted from the sensor data, achieving an impressive motion prediction accuracy of 95.83% for both linear and polynomial kernel functions. Furthermore, the research highlighted the superiority of the GC ink used in the printing process. This ink exhibited excellent printability and stability, enabling the creation of conductive paths essential for the sensor’s functionality. The multi-layered structure of the sensors, inspired by natural phenomena such as peacock feathers and butterfly wings, played a crucial role in their performance by enhancing their mechanical and sensing properties.

### 5.2. Disease Diagnosis and Management

Disease diagnosis and management involve the use of wearable sensors for the early detection, monitoring, and treatment of various diseases, such as diabetes, cardiovascular disorders, and cancer [121]. LbL-fabricated sensors have shown great potential for disease diagnosis and management applications due to their ability to detect specific biomarkers and bioreceptors with high sensitivity and selectivity. These sensors can be integrated into wearable platforms that are easy to use, painless, and economical, enabling the rapid and reliable diagnosis and management of diseases.

For example, Nag et al. [122] explored the potential of a novel sensor design for detecting volatile organic compounds (VOCs), crucial for early disease diagnosis, including cancer. Central to this innovation was the use of a LbL assembly method to functionalize CNT with polyhedral oligomeric silsesquioxanes (POSS), enhancing the sensor’s sensitivity and selectivity towards specific VOC biomarkers. The LbL technique allowed for precise control over the sensor surface properties, resulting in the ability to fine-tune the sensor’s response to various VOCs. Performance tests revealed that these sensors could detect VOC concentrations in ppm, with some configurations achieving sensitivity in ppb, showcasing their exceptional resolution. The study meticulously evaluated the sensors’ limit of detection, employing a signal-to-noise ratio (SNR) criterion for assessing efficiency at ppm to ppb levels of VOC presence. The sensors demonstrated a marked improvement in detecting and discriminating between different VOCs, attributed to the POSS-functionalized CNTs’ unique properties. The POSS molecules, when grafted onto the CNT surface, not only increased the thermal stability of the sensors but also significantly boosted their chemo-resistive behavior, enabling a more robust and reliable detection of VOC biomarkers. Chang et al. [123] developed an electrochemical biosensor utilizing a LbL assembly technique to quantitatively evaluate oxidative stress in tumor cells through the detection of H_2_O_2_ release. This biosensor incorporated rGO, Au NPs, and poly(toluidine blue O) (PTBO) films, leveraging the unique properties of each material to enhance sensitivity and selectivity towards H_2_O_2_. The study’s pivotal findings revealed that tumor cells emitted H_2_O_2_ at significantly higher levels than normal cells, underscoring the heightened oxidative stress within tumor environments. Specifically, the sensor’s performance was highlighted by its ability to detect H_2_O_2_ with high precision, demonstrating a substantial increase in current response when tumor cells were analyzed compared to normal cells. Ai [124] explored the application of LbL self-assembled polyelectrolyte capsules for enhancing magnetic resonance imaging (MRI) diagnostics. The study meticulously demonstrated how these LbL capsules, when embedded with paramagnetic metal-ligand complexes or superparamagnetic iron oxide nanoparticles, significantly improved MRI visibility and contrast, thus offering a promising avenue for more accurate and sensitive diagnostic imaging. Specifically, the capsules showcased an ability to provide detailed images by enhancing the contrast in areas of interest, which is critical for early detection and diagnosis of diseases. By integrating diagnostic and therapeutic functionalities, these LbL capsules pave the way for advancements in theranostics, offering a synergistic approach to disease management that combines diagnosis, therapy, and follow-up in one platform. Liu et al. [125] developed a sensitive electrochemical immunosensor for the detection of human immunoglobulin G (IgG). This innovative approach utilized a LbL assembly method to create a composite electrode interface, enhancing the electrochemical performance through the strategic application of PDDA (Figure 17). PDDA served a dual purpose: facilitating the layering process by forming a stable link between the negatively charged graphene and MWCTs, and significantly improving the electron transfer rate across the composite interface. The meticulous assembly resulted in an electrode that exhibited exceptional sensitivity and specificity in detecting human IgG, with a remarkable detection limit of 0.2 ng/mL. The immunosensor’s performance was further validated through clinical tests, where it demonstrated excellent reproducibility, stability, and selectivity. 

## 6. Challenges and Future Perspectives

Despite the significant advancements in the field of LbL-fabricated wearable sensors, there are still several challenges that need to be addressed to enable their widespread adoption and commercialization. These challenges include issues related to long-term stability and durability, biocompatibility and wearability, data acquisition and analysis, and large-scale manufacturing and commercialization. In this section, we will discuss these challenges in detail and provide insights into the future perspectives of LbL-fabricated wearable sensors.

One of the major challenges facing LbL-fabricated wearable sensors is their long-term stability and durability. Sweat composition varies among individuals and can be influenced by factors such as diet, hydration status, and physical activity. Moreover, the pH, ionic strength, and presence of interferents in sweat can fluctuate over time, which can affect the performance and reliability of the sensors. To address these issues, researchers are exploring the use of advanced materials, such as stimuli-responsive polymers and self-healing hydrogels, that can adapt to the changing sweat environment and maintain their functionality over extended periods. Additionally, the incorporation of protective coatings and encapsulation strategies can help to minimize the effects of sweat-induced degradation and biofouling on the sensor components. Another challenge facing LbL-fabricated wearable sensors is their biocompatibility and wearability. Wearable sensors are in direct contact with the skin for prolonged periods, which can cause irritation, inflammation, and allergic reactions. LbL-fabricated sensors are particularly prone to these issues due to the potential for leaching of the materials and the presence of residual chemicals from the fabrication process. To address this challenge, researchers are exploring the use of biocompatible and non-toxic materials, such as natural polymers, hydrogels, and ceramics, for the fabrication of LbL-fabricated sensors. Mass production at low cost is another critical consideration for the commercialization of LbL-fabricated wearable sensors. While the LbL technique offers precise control over the sensor architecture and composition, it is inherently a time-consuming and labor-intensive process. To overcome this bottleneck, researchers are investigating the use of automated and scalable fabrication methods, such as spray-assisted LbL assembly and roll-to-roll processing, which can significantly reduce the production time and cost. Furthermore, the use of low-cost and abundant materials, such as cellulose and chitosan, can help to minimize the overall sensor cost without compromising performance. The susceptibility of sensor data to environmental disturbances, such as temperature and humidity, is another important challenge that needs to be addressed. Variations in ambient conditions can affect the sensor response and lead to inaccurate or unreliable measurements. To mitigate these effects, researchers are developing multi-modal sensing platforms that can simultaneously monitor multiple parameters, such as temperature and humidity, alongside the target analyte. By incorporating reference sensors and compensation algorithms, it is possible to correct for environmental disturbances and improve the accuracy and reliability of the sensor data.

A third challenge facing LbL-fabricated wearable sensors is the acquisition, transmission, and analysis of the sensor data. Wearable sensors generate large amounts of data that need to be collected, processed, and analyzed in real-time to provide meaningful insights into the wearer’s health and well-being. However, the current data acquisition and transmission technologies, such as Bluetooth and Wi-Fi, are limited in terms of power consumption, data rate, and security. To address this challenge, researchers are exploring the use of low-power and high-bandwidth communication technologies, such as near-field communication (NFC) and ultra-wideband (UWB), for the wireless transmission of sensor data. Similarly, the use of edge computing and machine learning algorithms is being investigated for the real-time processing and analysis of sensor data on the wearable device itself, reducing the need for cloud computing and improving the privacy and security of the data. Finally, a significant challenge facing LbL-fabricated wearable sensors is their large-scale manufacturing and commercialization. LbL-fabricated sensors are currently limited to small-scale production in research labs, which is time-consuming, labor intensive, and expensive. To enable the widespread adoption and commercialization of LbL-fabricated sensors, there is a need for scalable and cost-effective manufacturing methods that can produce high-quality and reliable sensors in large quantities. To address this challenge, researchers are exploring the use of advanced manufacturing technologies, such as 3D printing, roll-to-roll processing, and automated LbL assembly, for the large-scale production of LbL-fabricated sensors.

Calibration is another critical aspect of translating raw sensor data into actionable information for health monitoring applications. Wearable sensors often require personalized calibration to account for inter-individual variability in sweat composition and skin physiology. However, this process can be time-consuming and inconvenient for users. To address this challenge, researchers are exploring the use of machine learning algorithms and artificial intelligence (AI) technologies to automate the calibration process and enable real-time data interpretation. By leveraging large datasets of sensor responses and corresponding physiological parameters, AI models can be trained to predict the target analyte concentrations directly from the raw sensor data, without the need for manual calibration. Moreover, AI can be used to identify patterns and correlations in the sensor data that may be indicative of specific health conditions or disease states, enabling early detection and intervention.

## 7. Conclusions

The recent advancements in polymer-assisted LbL fabrication of wearable sensors have opened up new possibilities for continuous health monitoring and disease diagnosis. LbL self-assembly has emerged as a powerful and versatile technique for creating conformal, flexible, and multi-functional films on various substrates, making it particularly suitable for fabricating wearable sensors. The incorporation of polymers, both natural and synthetic, has played a crucial role in enhancing the performance, stability, and biocompatibility of these sensors. Natural polymers, such as cellulose, chitosan, and silk fibroin, offer unique properties like biodegradability and biocompatibility, while synthetic polymers, including polyelectrolytes, conductive polymers, and stimuli-responsive polymers, provide tunability and functionality. LbL-fabricated wearable sensors have demonstrated exceptional capabilities in detecting and monitoring various physical, chemical, and biological parameters, enabling their applications in continuous health monitoring, disease diagnosis, and management. Despite the significant progress, challenges related to long-term stability, biocompatibility, data acquisition, and large-scale manufacturing still need to be addressed to realize the full potential of these sensors. Future research should focus on developing novel materials, optimizing the fabrication processes, and integrating advanced data processing and communication technologies to create reliable, cost-effective, and user-friendly wearable sensors. With continued advancements in polymer-assisted LbL fabrication and related fields, wearable sensors are poised to revolutionize personalized healthcare and improve the quality of life for individuals worldwide. 

## Figures and Tables

**Figure 1 sensors-24-02903-f001:**
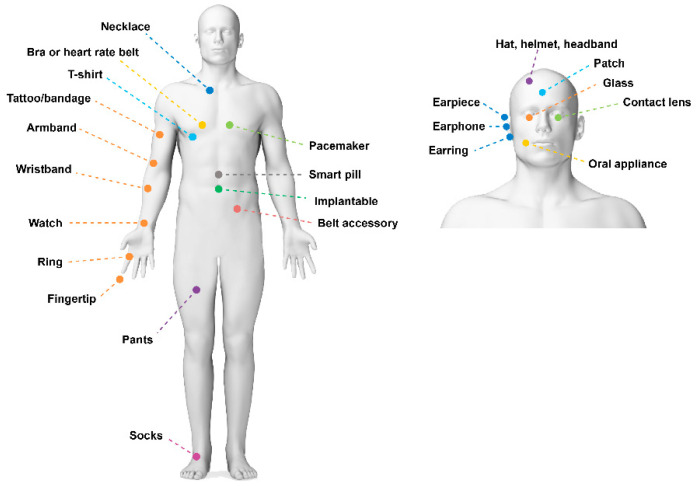
A schematic of different parts of the body that can be used for health sensing monitoring.

**Figure 2 sensors-24-02903-f002:**
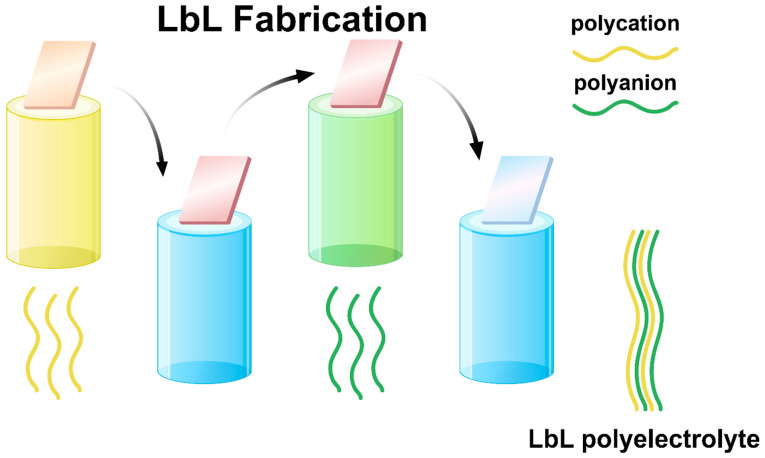
A schematic illustration of the alternate adsorption of the polyelectrolyte species to produce a multi-layered structure.

**Figure 3 sensors-24-02903-f003:**
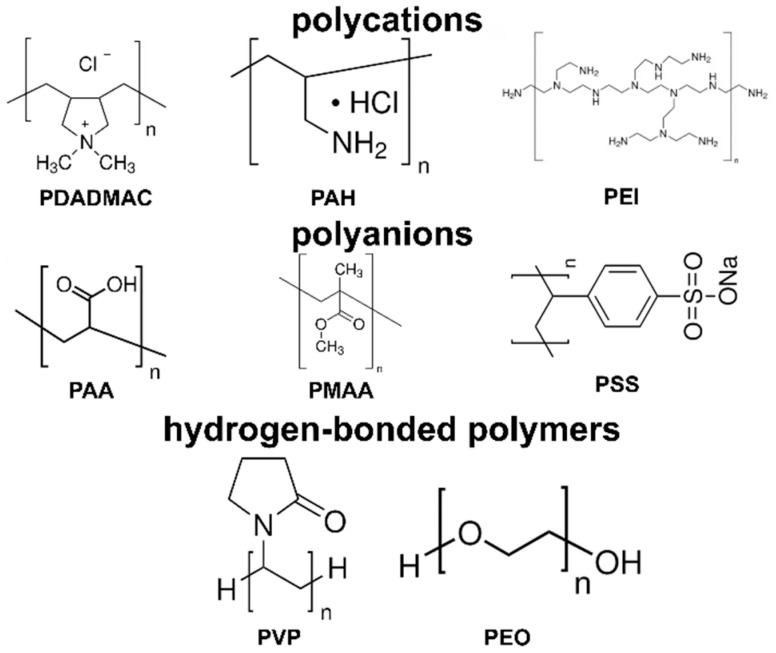
Common polymers used for LbL assembly.

**Figure 4 sensors-24-02903-f004:**
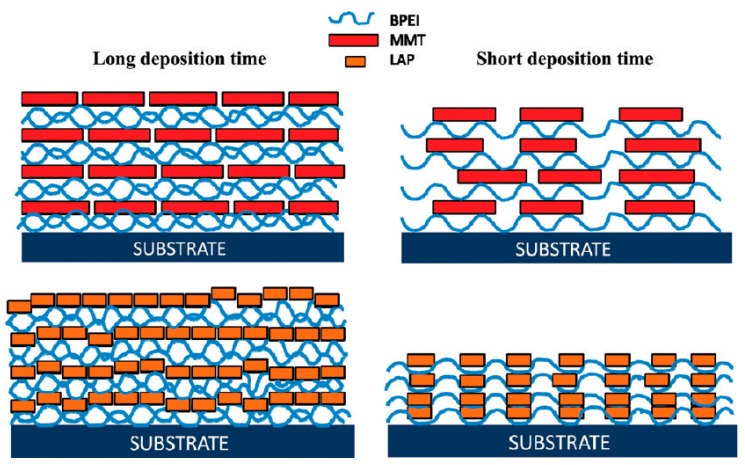
Film growth mechanism for BPEI/MMT and BPEI/LAP with different deposition times [52].

**Figure 5 sensors-24-02903-f005:**
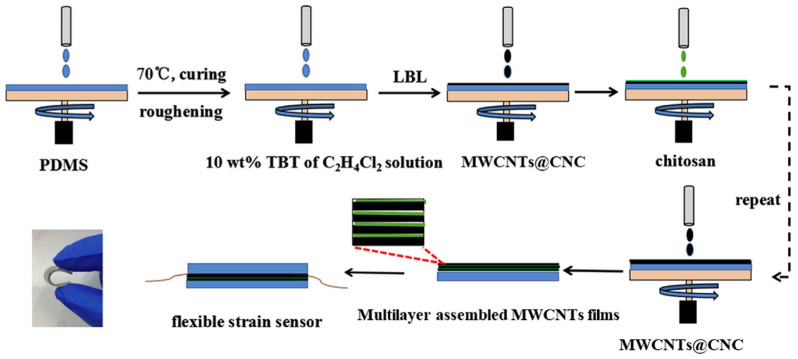
Schematic illustration of the fabrication of the multi-layer assembled MWCNTs@CNC-PDMS nanocomposites and photograph of the strain sensor being folded [65].

**Figure 6 sensors-24-02903-f006:**
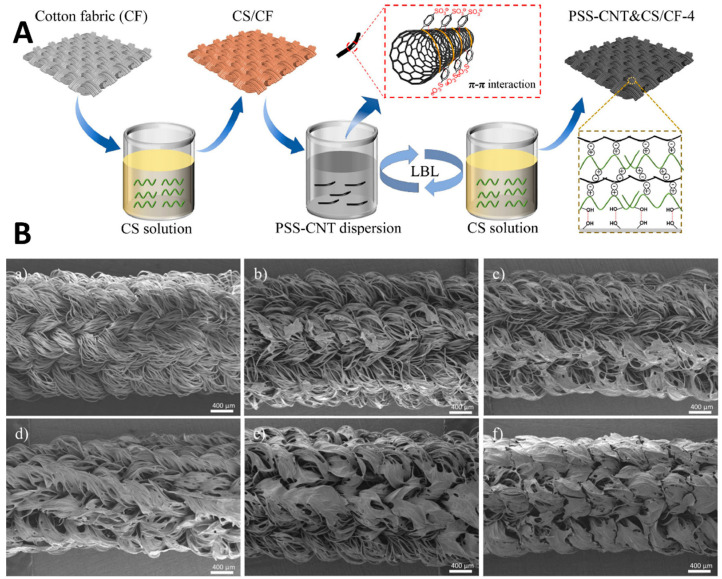
(**A**) LBL assembly of CS and PSS-CNT on CF to fabricate the FCCF [70]. (**B**) SEM images of (**a**) control fiber, (**b**) (rGO)_6_@PF, (**c**) (rGO@CS)_1_@PF, (**d**) (rGO@CS)_6_@PF, (**e**) (rGO@CS_1_)_6_@PF, (**f**) (rGO@CS_3_)_6_@PF [72].

**Figure 7 sensors-24-02903-f007:**
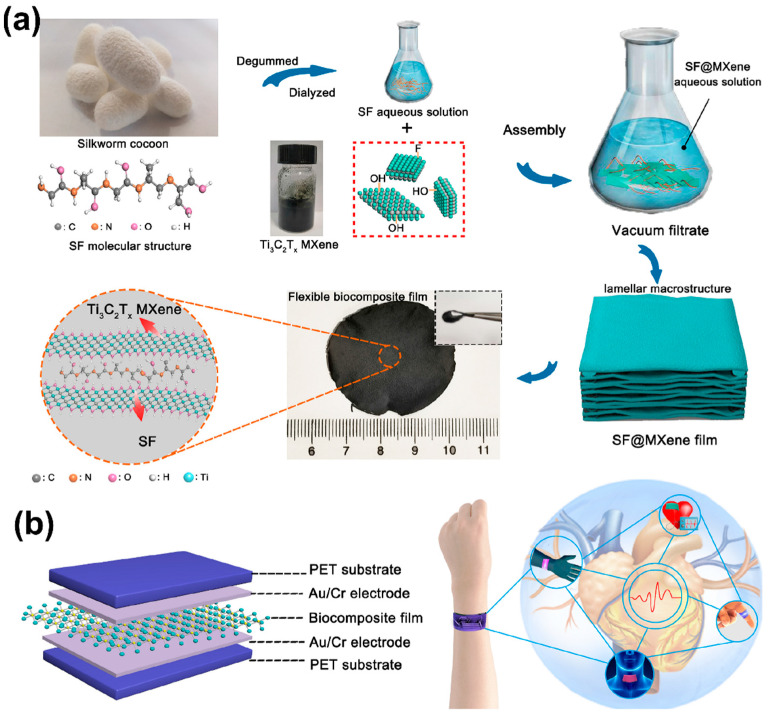
(**a**) Illustration of preparation of the SF@MXene biocomposite film and flexible pressure sensor. (**b**) Schematic illustration of SF@Mxene biocomposite film-based flexible pressure sensor and its application in human health detection [77].

**Figure 8 sensors-24-02903-f008:**
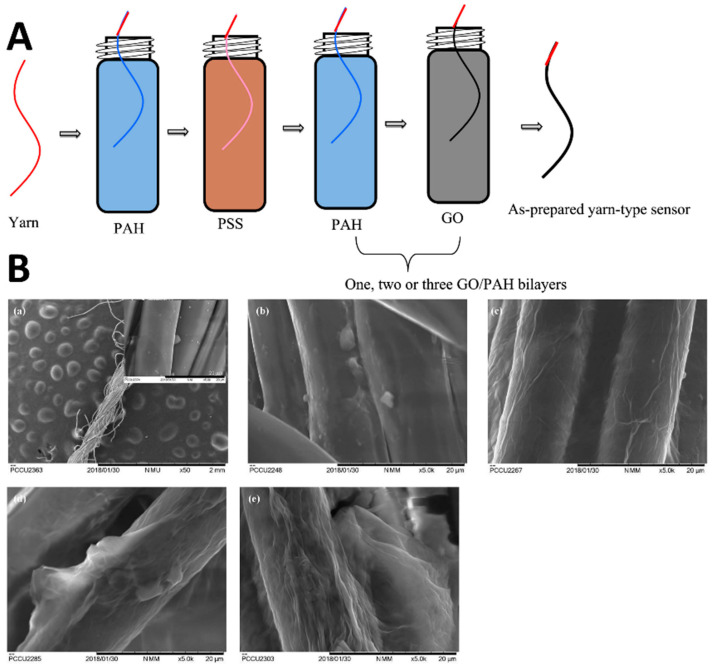
(**A**) Schematic diagram of GO multi-layer thin films assembled on a single yarn. (**B**) SEM images of (**a**) bare yarn (inset: enlarged image), (**b**) (PSS/PAH)/yarn, (**c**) (GO/PAH)_1_(PSS/PAH)/yarn, (**d**) (GO/PAH)_2_(PSS/PAH)/yarn and (**e**) (GO/PAH)_3_(PSS/PAH)/yarn [80].

**Figure 9 sensors-24-02903-f009:**
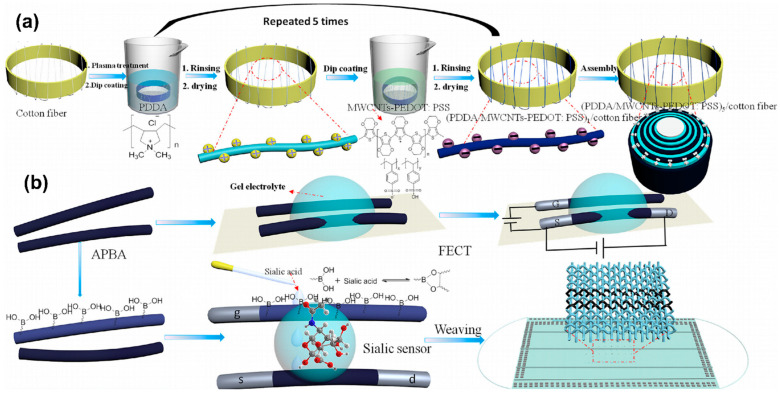
(**a**) Schematic diagram of LBL assembly of composites conductive fiber. (**b**) Preparation of FECT and FECT based sialic acid sensor [86].

**Figure 10 sensors-24-02903-f010:**
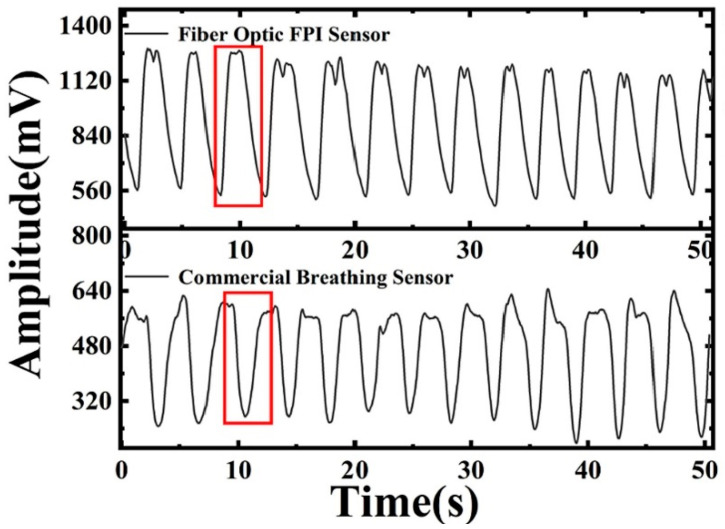
Partial enlarged view of the respiratory response monitored by the fabricated FPI sensor [95].

**Figure 11 sensors-24-02903-f011:**
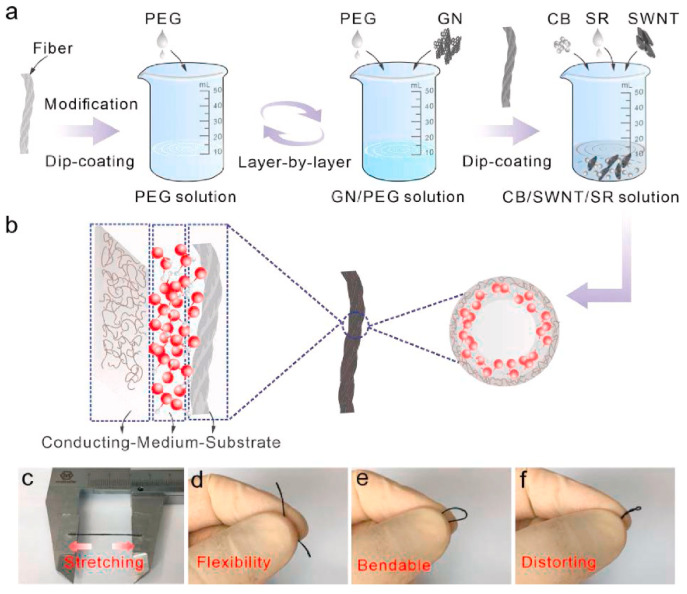
(**a**) Fabrication processes of biomimetic flexible strain sensor. (**b**) The sensing framework of conducting-medium-substrate for the strain sensor. (**c**–**f**) Optical graphs of the strain sensor, demonstrating high flexibility, good bendability [98].

**Figure 12 sensors-24-02903-f012:**
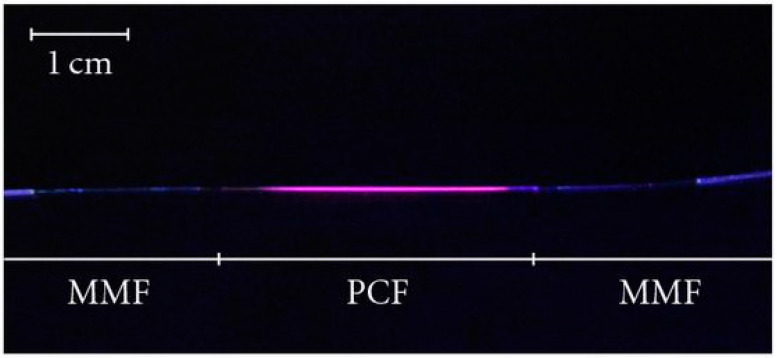
Picture of the LMA20 segment spliced to the MMF fibers under UV illumination [102].

**Figure 13 sensors-24-02903-f013:**
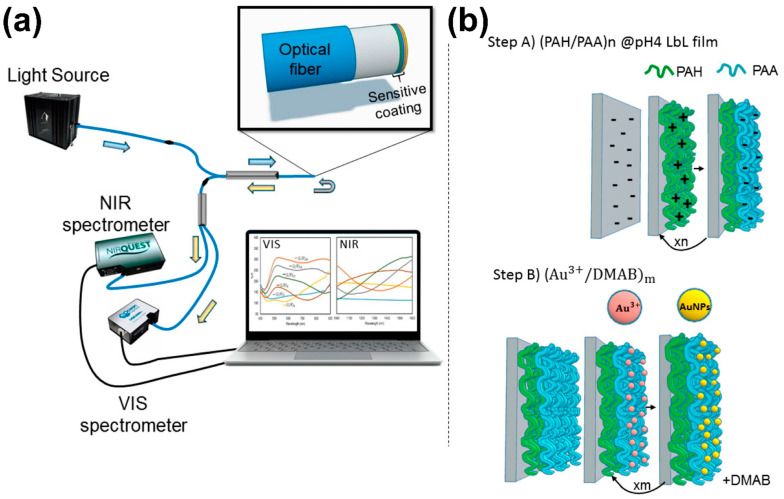
(**a**) Experimental setup and (**b**) schematic illustration of the LbL nano-assembly built-up and a further ISS of the AuNPs into the previously created LbL films [107].

**Figure 14 sensors-24-02903-f014:**
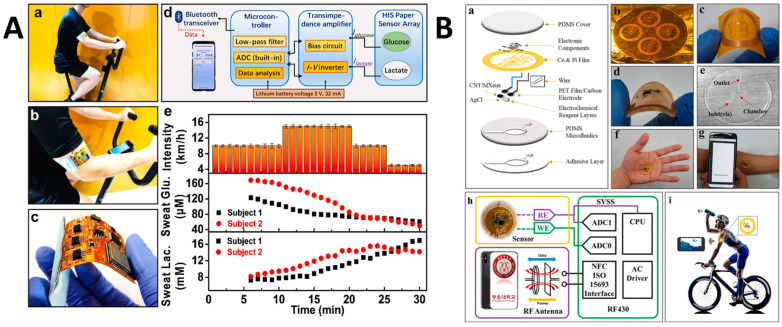
(**A**) Real-time on-body evaluation of the paper-based sweat monitoring system. (**a**) The photograph of the sweat analysis with the paper-based system mounted on the arm during physical activity. (**b**) The detection process with the wireless patch to receive the detection data and display the results. (**c**) Detailed photograph of the flexible circuit board. (**d**) Illustration of the analysis system. (**e**) The real-time concentration curve of sweat glucose (Glu.) and sweat lactate (Lac.) of volunteers during cycling with corresponding exercise intensities [110]. (**B**) (**a**) Schematic illustrating the exploded view of the wireless, battery-free, flexible, microfluidic/electronic system. (**b**,**c**) The lithography and etching process is used to make the flexible antenna and circuits. (**d**) Encapsulation of flexible circuits with PDMS. (**e**) The microfluidic system. (**f**) A very light and small-sized circular patch sensor system. (**g**) Wireless NFC sensor and smartphone communication in real time. (**h**) Electronics design, measurement strategy, and complete system. (**i**) Measuring the electrolyte losses from athletes’ sweat could help health professionals plan personalized water and electrolyte replacement strategies [111].

**Figure 15 sensors-24-02903-f015:**
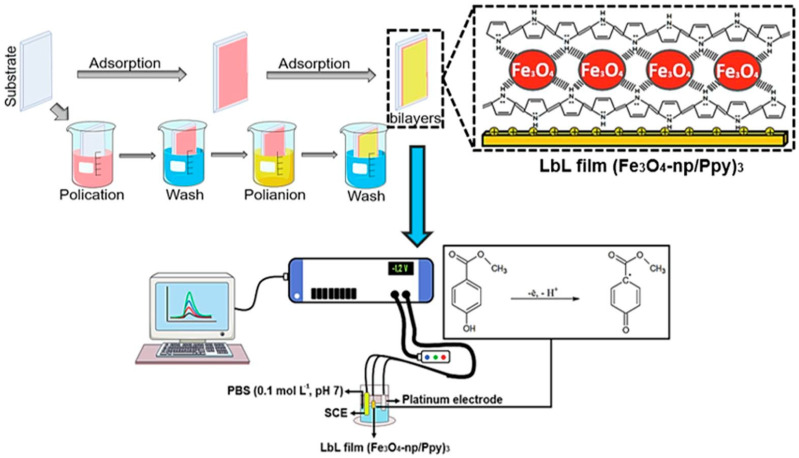
LbL films of magnetite nanoparticles and Ppy towards methylparaben electrochemical detection [112].

**Figure 16 sensors-24-02903-f016:**
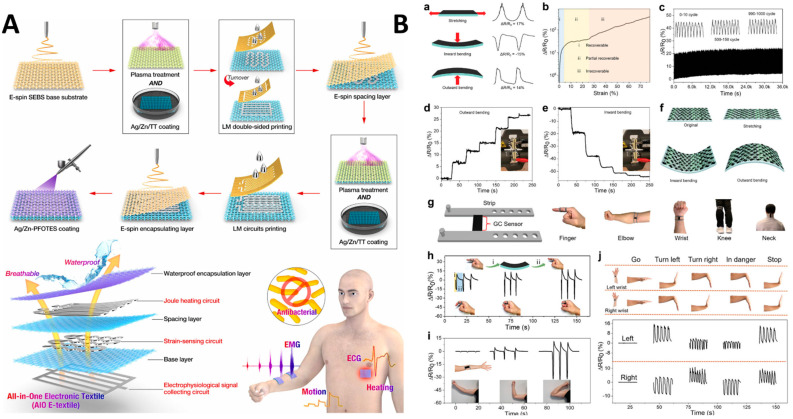
(**A**) Schematic for the fabrication of the AIO E-textile. Illustration for the detailed configuration of designed AIO E-textile with integrated functionalities [119]. (**B**) (**a**) Detection signals of the GC sensor under different deformations; (**b**) change in normalized resistance against strain; (**c**) cyclic test results of 1000 stretch–release cycles of the GC sensor tensile bar with 5% elongation; the GC sensor monitors the response signals of (**d**) outward bending and (**e**) inward bending; (**f**) schematic illustration of the changes in the conductive layer under external forces; (**g**) the design and applications of the printed GC sensors; output signals of the GC sensor collected at the bending of (**h**) finger and (**i**) elbow; (**j**) discrimination of emergency signs based on two GC sensors worn on both wrists [120].

**Figure 17 sensors-24-02903-f017:**
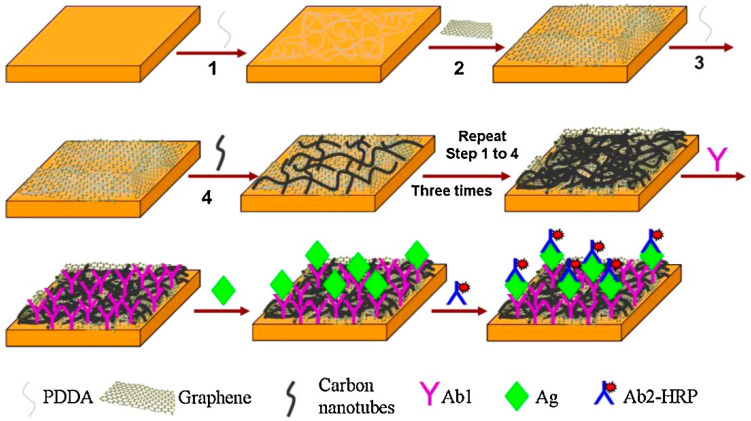
Schematic illustration of the fabrication of graphene–carbon nanotube electrode interface and the construction of the sandwich-type electrochemical immunosensor [125].

**Table 1 sensors-24-02903-t001:** Physical sensors prepared via LbL method and their performance.

Sensor Component	Performance	Reference
**Strain and Pressure Sensors**		
Polymer used: PDDA; PSS Conductive materials: graphene	Gauge factor: 78.13	[97]
Polymer used: polyurethane; polyethylene glycol Conductive materials: graphene nanoplatelets; carbon black; silicon rubber	Gauge factor: 2.0 at 50% strain Response time: 60 ms Durability: 3000 cycles	[98]
Polymer used: PDMS Conductive material: CNT	Gauge factor: 35.75 Stretchability: up to 45%	[99]
Polymer used: polyurethane yarn; PDMS Conductive material: graphene nanosheets; thin gold film	Gauge factor: 661.59 Durability: applied strain of 50% for 10,000 cycles	[100]
Polymer used: polyaniline fiber Nanocomposite material: MXene sheets	Gauge factor: 2369.1 Stretchability: up to 80%	[101]
**Temperature Sensors**		
Polymer used: PDDA; PAA Conductive materials: CdSe quantum dots	Detection range: -40–70 °C	[102]
Polymer used: polyacrylonitrile Conductive materials: GO	Detection range: 20–100 °C Sensitivity: −0.4%/°C	[103]
Polymer used: polyacrylonitrile	Detection range: 15–50 °C	[104]
**Optical Sensors**		
Polymer used: PAH Counterpart material: BY	pH range: 6.80 to 9.00 Wavelength shift: 4.65 nm per 0.2 pH units	[105]
Polymer used: PAH Counterpart material: SiO_2_	pH range: 6.00 to 9.00	[106]
Polymer used: PAH; PAA Counterpart material: AuNPs	RH range: 20 to 80%	[107]

## Data Availability

Data are contained within the article.

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
