# Peer review of "Advancements in Polymer-Assisted Layer-by-Layer Fabrication of Wearable Sensors for Health Monitoring"

_sensors, 2024, doi:10.3390/s24092903_

Round 1

Reviewer 1 Report

Comments and Suggestions for Authors

The manuscript is dedicated to review of recent polymer-assisted layer-by-layer fabrication techniques for wearable sensors and their applications where different aspects including selections of polymeric materials and capabilities in detecting and monitoring various physical, chemical, and biological features are discussed. The manuscript further summarizes the current research status for various health monitoring and sensing devices, and addresses the limitations of current research, further providing suggestions on its development direction in the future. The manuscript is well-articulated, clear, and of interest; however, a few issues may need further clarification.

  1. In Fig. 2 (page 4), the schematic illustration on polymer LbL technique is unclear.  
  2. In p4, the paragraph addressing PH and temperature is very similar to the paragraph 3&4 in p6.
  3. In Fig. 3 (page 5), it is suggested that the positively-charged polymers and negatively-charged polymers for LbL should be categorized. 
  4. The caption for Fig. 5B (page 10) may need to add a proper citation(s).  
  5. Table 1 is seemingly a bit like a rough note. 

Reviewer 2 Report

Comments and Suggestions for Authors

The manuscript comprehensively discusses the research advancements in wearable sensors for health monitoring, fabricated using polymer layer-by-layer (LbL) techniques, and provides a thoughtful discussion and outlook. While the choice of topic is commendable, the readability of the paper could be improved. Below are several suggestions intended to enhance the quality of the manuscript:

In Section 2.3, "Factors Affecting LbL Film Properties", the paper discusses the main factors influencing LbL film performance and cites relevant studies. It is recommended to include illustrative images to enhance readability. Similarly, sections "3.2.3 Stimuli-responsive Polymers" and "4.1.2 Temperature Sensors" would also benefit from additional figures.

Figure 6 contains two images but lacks individual labels for clarification. The same issue is present in Figures 8 and 10; adding labels would improve clarity and reference efficiency.

The font size and style in Figures 5, 7, 9, and 12 should be uniform. Additionally, the font type used within these figures should also be standardized to maintain consistency throughout the document.

Regarding Section 4, "4.1 Physical Sensors", summarizing the examples mentioned in a table would provide a clearer overview, and the content of "Table 1" could be incorporated into this summary. The narrative within "Table 1" appears redundant and should be streamlined. Moreover, the comparison parameters in the "performance" section are inconsistent and should be unified. If feasible, including detailed data for these parameters would be beneficial.

The section "4.2 Chemical Sensors" seems underdeveloped compared to "4.1 Physical Sensors". My understanding after reading this section is that it does not adequately focus on the LbL-fabricated "Chemical Sensors", resembling more the content of "5. Applications of LbL-Fabricated Wearable Sensors". It is advised to expand on the detailed description of LbL-fabricated "Chemical Sensors", including additional photos and detailed parameters. Moreover, there is significant recent exemplary work in wearable electrochemical and non-electrochemical sensors that could be discussed, such as in https://doi.org/10.1021/acssensors.2c02016.

Section 6 requires a more in-depth discussion. The authors should assess the current research level of LbL-fabricated wearable sensors and identify the bottlenecks that need to be overcome before commercial application. As mentioned, long-term stability and durability are universal challenges across all sensors. What makes wearable sweat sensors uniquely challenging in this regard? Additionally, considerations for mass production at low cost, susceptibility of sensor data to environmental disturbances (such as temperature and humidity), and the need for calibration to translate raw data into actionable information should be addressed. With the advancement of AI technologies, the possibility of automating these intermediary steps could be explored.

Round 2

Reviewer 2 Report

Comments and Suggestions for Authors

the manuscript has been improved.